# The Effect of Fuel Cell and Battery Size on Efficiency and Cell Lifetime for an L7e Fuel Cell Hybrid Vehicle

**Tom Fletcher ***  and **Kambiz Ebrahimi**

School of Aeronautical and Automotive, Chemical and Materials Engineering (AACME),
Loughborough University, Loughborough LE11 3AP, UK; k.ebrahimi@lboro.ac.uk
* Correspondence: T.P.Fletcher@lboro.ac.uk

**Abstract:** The size of the fuel cell and battery of a Fuel Cell Hybrid Electric Vehicle (FCHEV) will heavily affect the overall performance of the vehicle, its fuel economy, driveability, and the rates of fuel cell degradation observed. An undersized fuel cell may experience accelerated ageing of the fuel cell membrane and catalyst due to excessive heat and transient loading. This work describes a multi-objective design exploration exercise of fuel cell size and battery capacity comparing hydrogen fuel consumption, fuel cell lifetime, vehicle mass and running cost. For each system design considered, an individually optimised Energy Management Strategy (EMS) has been generated using Stochastic Dynamic Programming (SDP) in order to prevent bias to the results due to the control strategy. It has been found that the objectives of fuel efficiency, lifetime and running cost are largely complimentary, but degradation and running costs are much more sensitive to design changes than fuel efficiency and therefore should be included in any optimisation. Additionally, due to the expense of the fuel cell, combined with the dominating effect of start/stop cycling degradation, the optimal design from an overall running cost perspective is slightly downsized from one which is optimised purely for high efficiency.

**Keywords:** fuel cell sizing; Stochastic Dynamic Programming; hybrid vehicle; hydrogen fuel cell; fuel cell durability; fuel cell cost reduction

## 1. Introduction

Fuel Cell Hybrid Electric Vehicles (FCHEVs) represent a promising low carbon personal transportation alternative for the future [1]; however, their mainstream adoption is currently prohibited by several issues including their high cost to manufacture, poor reliability and the limitations of on-board hydrogen storage. Fortunately, many of these issues can be combated at a system level to optimise the overall vehicle design.

Primarily, a larger fuel cell will be able to provide a higher maximum power improving the vehicle's straight-line performance (if not already limited by the electric motor). However, there are a few secondary benefits of using a fuel cell with a higher peak power. Firstly, the increased power overhead will likely improve the response of the vehicle, improving driveability concerns [2] with respect to power availability in the event of a sudden potential acceleration [3]. Secondly, the operating power will likely be shifted towards a more efficient operating region because fuel cells are typically most efficient at just 30% of their maximum power [4]. Finally, by operating the fuel cell further from its maximum power, the lifetime of the fuel cell can also be extended. This is not only due to thermal degradation observed at high power, but also due to higher power availability for the avoidance of transient loading, a degradation mechanism which affects the catalyst and membrane in particular and known to be very significant for transport applications [5–9].

However, there are also a few downsides to using a high-powered fuel cell. The most obvious of these is the increased cost and mass associated with a larger stack [10]. A larger fuel cell stack has the potential to significantly affect the overall mass of the vehicle and as a result, negatively impact its straight-line performance and fuel economy. The increased mass will also result in an increase to the load on the stack, potentially resulting in higher rates of degradation. Similarly, the greater cost of the stack will not only increase the upfront cost of the vehicle, but also proportionally increase the cost of replacement and therefore the proportional cost of any degradation.

The trade-off regarding the size of battery pack is very similar. A larger battery will provide a higher peak power and be able to absorb more braking energy [2], potentially improving driveability concerns and the straight-line performance of the vehicle. A larger pack will also provide a larger energy buffer, allowing the fuel cell to reside in its most efficient operating region. However, a large battery pack will also result in its own cost and weight penalties.

The objective of this work was to investigate the effect of varying fuel cell stack size and battery pack capacity on these four coupled vehicle characteristics: (1) fuel consumption, (2) fuel cell lifetime, (3) range, (4) fuel cell operating cost.

### 1.1. Background

This work concerns the design optimisation of a low-power FCHEV designed for light personal transport (taxi services) and mail delivery on a university campus where the maximum speed limit is 20 mph. It has been found that the diesel vehicles used for campus-only services were responsible for emitting approximately 400 tons of $CO_2$ per year [11]. This was attributed to the low speed and intermittent usage patterns experienced on campus, which are well outside the typical duty cycles for traditional road vehicles.

In order to combat this issue, five lightweight FCHEVs (Microcab H4s) were produced under the EU L7e ("Heavy Quadricycle") classification. L7e classification offers a number of benefits, such as a small size, low mass, and lower development cost, reducing many of the obstructions to current fuel cell technology mentioned previously. In addition, the localised usage pattern means that just a single hydrogen refueller is required to be installed, eliminating the issue of infrastructure. The high efficiency and zero emission capabilities of FCHEVs makes them ideal for this type of usage and for similar low power, urban use cases, such as forklifts [12].

The Microcab H4 FCHEVs accumulated more than 4000km of real-world usage over a period of 2 years between 2009 and 2011; however, analysis of their usage [13] showed just 18% tank-to-wheel efficiency. This was below expectations, and significantly below that of other fuel cell vehicle concepts at the time. It was determined that significant gains could be made by optimisation of the control strategy and a system level re-design of the sizing of various powertrain components; in particular, the 1.2 kW fuel cell which was too small to maintain the battery State of Charge (SoC) over the typical usage cycles [14]. Previous work by the authors [15] has shown that an up-sized 4.8 kW fuel cell eliminates this problem and also, by using optimal Stochastic Dynamic Programming (SDP) control, the fuel economy could be improved by around 27% when compared to the original vehicle. However, it was also shown that the 4.8 kW fuel cell exhibited high degradation costs due to on/off cycling and was most likely oversized for the duty cycle.

It is therefore desirable to determine how the sizing of both fuel cell and battery pack affect the performance, efficiency, and use-ability characteristics of an FCHEV within the constraints of the L7e classification.

### 1.2. Prior Literature

Before a component sizing exercise can take place, the designers must first determine which components will be used in the hybrid powertrain. Although it is possible to use a fuel cell as the sole power source on the vehicle, hybridisation with batteries is much more common [16–27]. This allows

transient loading to be absorbed by the battery, increasing the lifetime of the fuel cell, and allowing it to operate more efficiently [28–30].

Hybridisation with supercapacitors (or ultracapacitors) has also been suggested [31–33]. Supercapacitors offer the advantage that they have a much higher power density than batteries making them more efficient at absorbing regenerative braking energy. Although they are capable of absorbing transient loading effectively, supercapacitors tend to have a lower energy density than batteries and therefore combining them with batteries to make a dual energy store has also been popular in recent years [34–37], rather than using them without a battery pack.

There is no clear consensus in the literature as to which design is best, but there has been a number of studies to investigate the issue. In general, most component sizing exercises for fuel cell vehicles tend to focus on fuel cell/battery hybrids [4,38,39]. This is generally because a battery is usually required to store enough energy for fuel cell start-up [40]. Bauman and Kazerani [41] and others [40,42] have found that the inclusion of ultracapacitors with a battery pack was marginal, extending the lifetime of the battery and improving the fuel economy, but also increasing the cost of the system.

There are numerous approaches for determining the optimal selection of component sizes. Ang et al. [28] separate these into three categories: parametric studies, single-objective optimisations and multi-objective optimisations. Parametric studies are often either used to select between discrete available component options by commercial entities [39] or to describe the trends seen as certain input parameters are varied [43]. Work involving optimisation techniques on the other hand tend to focus on the algorithms used and their speed and effectiveness at identifying a globally optimal solution [38,44].

Many studies focus on either system design [45,46] or Energy Management Strategy (EMS) [47–51], when in-fact, these two aspects are strongly coupled and therefore should be considered concurrently [35,38,43,52]. In particular, Rousseau et al. [4] mentioned the energy storage requirement of a vehicle was significantly impacted by changes to the parameters used in their rule-based EMS control strategy.

Deterministic Dynamic Programming (DDP) and SDP-based strategies eliminate this problem by individually optimising the control strategy based on the evaluated system design, meaning optimal potential performance can evaluated for each design. Sinoquet et al. [43] have presented a component sizing parametric study using a DDP-derived strategy and Kim and Peng [38] have presented an optimisation problem used to choose the component sizing to maximise the fuel economy using SDP and "pseudo-SDP" strategies.

*1.3. Contributions*

Since the work by Kim and Peng [38] in 2007 and by Sinoquet et al. [43] in 2011, computer processing power has improved significantly. Therefore, for this work, it has been possible to use a 40-core high performance workstation to derive stochastically optimal SDP strategies for each individual design rather than using *pseudo*-SDP or DDP-*derived* strategies. This eliminates any bias to the results as a result of the control strategy.

This work is also believed to be novel in that a mathematical model of the degradation previously developed by the authors [15] is included in these calculations. This enables this work to numerically compare how varying sizes of fuel cell and battery affect the efficiency, range, durability, and the operating cost. These criteria have been selected based their particular importance to zero carbon transportation, due to the high cost of fuel cells [53–55], and their below target reliability [5,56–58], which are two critical obstructions to the mainstream commercialization of FCHEVs.

Although it is generally understood that over-sizing the fuel cell stack and/or battery pack will lead to generally improved fuel cell lifetime [6–8], no attempt has been found to quantify this benefit and compare the potential running cost saving to the increased system mass and upfront cost in order to determine globally optimal fuel cell stack and battery pack sizes under real-world driving conditions.

Finally, in contrast to other research in this area, this work focusses on L7e classification vehicles. As mentioned previously, these vehicles have significantly reduced development costs and their typical usage patterns are highly suited for FCHEV technology making them the ideal platform for current FCHEV technology development.

## 2. Modelling

A full mathematical description of the model is available in previous work by the authors [15]; however, for the purpose of completeness, an outline is provided below.

### 2.1. Vehicle

The Microcab H4 is a low-speed passenger car developed specifically for university campus duty cycles. It uses a Ballard Nexa 1200 W PEM fuel cell alongside a 2100 Wh lead acid battery to power a 15 kW electric motor up to a maximum speed of around 30 mph. The rear wheels are directly driven using a brushed DC motor which is connected in parallel to the battery pack and a DC/DC converter driven by the fuel cell, see Figure 1a. The fuel cell output power can be controlled using an output current demand from this DC/DC converter.

A reduced-order backward-facing model (Figure 1b) of the vehicle has been developed for this project using Simulink. It is extremely important to minimise the complexity of the model because the model needs to be iterated around 2.5 million times. A backward facing model using unidirectional calculations based on power flow between components and minimizing feedback loops has been found to maximise the computational efficiency and perform within reasonable accuracy limits as long as the inputs and demands of the model are within the performance limits of the actual vehicle. As a result, this type of model is popular in the literature for SDP optimisation [59].

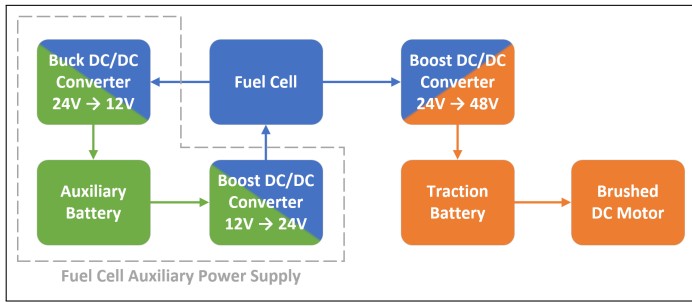
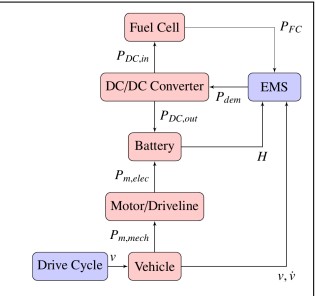

(**a**) Microcab Electrical Powertrain Schematic  (**b**) Backward Facing Model Outline

**Figure 1.** Vehicle model.

For the purpose of evaluation of different size battery packs, the specific energy of the existing lead acid battery is used, which is around 30 Wh/kg. Similarly, the specific power of the current fuel cell (0.1 kW/kg) is used to estimate the fuel cell mass.

Cost estimates for both hydrogen and fuel cells vary considerably depending on a number of factors such as production volumes, raw material costs and in the case of hydrogen, the production source. This can be carbon intensive from reforming natural gas (<$2/kg [12]), low-carbon reforming using carbon capture and storage (CCS) ($2–4/kg [12]) or zero carbon using electrolysis and renewable electricity sources (>$6/kg [12]). For the purpose of calculating the running cost, the cost of the fuel cell is assumed to be $50/kW [15,55,60–62], and the cost of fuel is assumed to be $3/kg [12,15,55,61]. These figures represent the medium-term (3–5 year) estimates for mass produced fuel cells and low-carbon hydrogen. Finally, it should be noted that the absolute costs of the hydrogen and fuel cell are of less importance than their relative values for weighting the degradation against fuel consumption.

## 2.2. Fuel Cell Degradation

The loading pattern on the fuel cell can significantly affect the ageing of the stack. Firstly, prolonged high-power usage can cause fuel starvation, due limitations of reactant supply, which may lead to oxidation of the carbon catalyst support material and detachment of the catalyst [8,63]. In addition, high power usage results in additional thermal stress which can cause sintering of the catalytic particles. Both of these degradation mechanisms reduce the Electro-Chemical Active Surface Area (ECASA), resulting in a drop in fuel cell performance. Excess heat may also lead to a reduction of the protonic conductivity and drying of the membrane [7,8]. This increases the electrical impedance of the cell which may lead to further heating.

Conversely, low current densities, resulting in high cathode potentials, can increase the concentration of surface oxides on the catalyst particles. The particles will tend to agglomerate when subsequently reduced [58] causing further, permanent, ECASA reduction.

Transient loading and fuel cell start/stop also results in significant degradation. This is believed to be due to the internal operating conditions of the fuel cell exceeding the transient control capabilities of the stack ancillaries designed to manage temperature, humidity, and reactant supply. As a result, localised fuel starvation, thermal loading, and flooding of the fuel cell trigger degradation mechanisms mentioned above due to inhomogeneous internal conditions.

For the purposes of modelling the fuel cell degradation, we assume the following loss of performance-based figures (Table 1), based on the manufacturer's datasheet and similar work in the literature [55] .

**Table 1.** Fuel cell performance loss [15].

| Operating Conditions | Degradation Rate |
| --- | --- |
| Low Power (<10%) | 10.17 µV/h |
| High Power (>80%) | 11.74 µV/h |
| Transient Loading | 0.0441 µV/Δ kW |
| Start/Stop | 23.91 µV/cycle |

## 3. Methodology

Rather than performing an optimisation exercise, it has been chosen to explore the design space over a range of fuel cell sizes and battery capacities. This was decided because it is not known how the multiple objectives may compete with each other and whether or not there may be multiple local optima corresponding to differing levels of hybridisation and the weighting of the objective criteria. By performing a design of experiment exercise over the design space, the response of each of the objectives can be individually quantified.

### 3.1. Design of Experiment

It has been found [13,14] that the 1.2 kW fuel cell is underpowered for its application, but a 4.8 kW fuel cell would likely be overpowered for campus usage [15]. This suggests that the optimal size of fuel cell is likely somewhere between these values, and therefore it has been decided to explore fuel cells with a maximum power of between 1.2 kW and 4.8 kW.

The 2.1 kWh battery pack in the Microcab H4 has been found to be capable of balancing the load on the fuel cell reasonably effectively. However, only approximately 6% SoC range was utilised by the SDP controller. Therefore, a smaller battery may be just as effective whilst also reducing the mass of the vehicle. Equally, it may be possible that a larger battery may allow better load balancing reducing the running cost of the vehicle despite the additional mass. For the initial design exploration, it has been decided to examine a range of battery capacities ranging from half to twice the original battery pack capacity.

Between these extreme values, a bisector search algorithm has been used to progressively increase the resolution of the results for a fixed period of time. This will give the highest resolution of results to produce a clear picture of the design space in a pre-determined period of time.

*3.2. Overall Process*

The process by which the results are gathered is described below and shown in Figure 2:

1.  The first step of the process is to produce a Markov chain model of the real-world usage pattern based on logged data. This model provides the likelihood of subsequent vehicle acceleration given its current speed and acceleration and provides the input probabilities for simulation in the vehicle model. It does not include any vehicle-specific characteristics; therefore, it is independent of the vehicle system design and only needs to be calculated once.
2.  The vehicle model is then parametrized using the desired fuel cell and battery size. Each possible combination of speed and acceleration (from the Markov model) is simulated for each possible control action in order to calculate the subsequent state of the vehicle and the resultant cost of the performed action.
3.  The simulation results provide a table of costs and probabilities which describe a Markov Decision Problem (MDP) and can then be solved using SDP to generate the optimal control strategy for that vehicle design.
4.  The resultant EMS is then used to simulate the vehicle completing a number of logged journeys in order to estimate the real-world fuel consumption and fuel cell degradation.
5.  Steps 2–4 are then repeated for each desired vehicle design and the results compared. Each design is optimised and evaluated using parallel processing, meaning that up to 40 designs can be processed concurrently using our hardware.

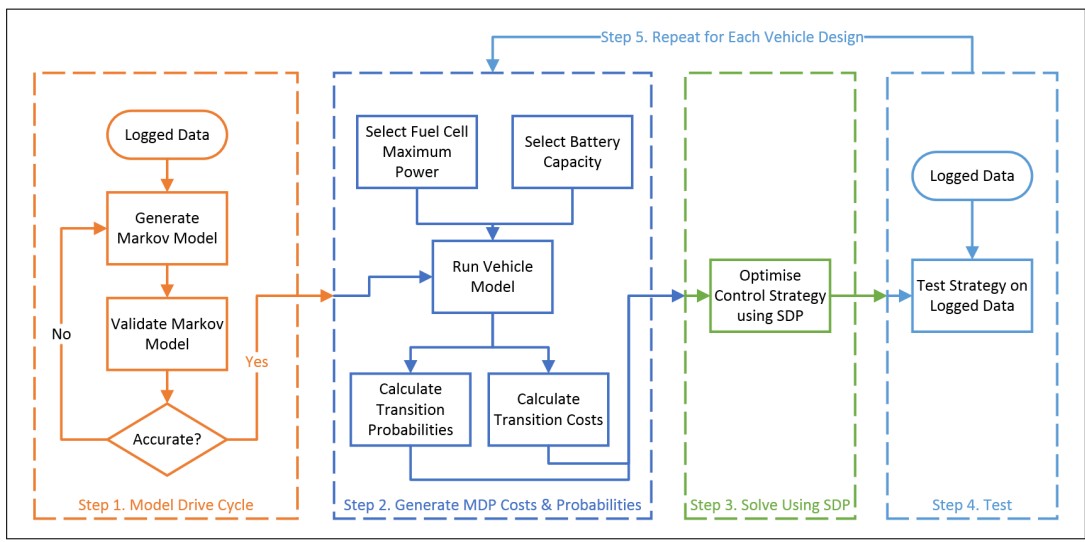

**Figure 2.** Methodology.

*3.3. Control Strategy*

The control strategy is based on Stochastic Dynamic Programming (SDP). SDP optimisation requires the duty cycle of the vehicle to be defined as a statistical model which describes the likelihood (and associated cost) of the vehicle transitioning from one state to another dependent on the action of the EMS. The state of the vehicle is defined by its current velocity, $v$, acceleration, $a$, the battery state of charge, SoC and the current fuel cell output power, $P_{FC}$:

$$\mathbf{S} = \mathbf{S}(v, a, \text{SoC}, P_{FC}(t)) \tag{1}$$

The action of the EMS corresponds to the subsequent demanded fuel cell power:

$$P_{FC}(t+1) = \{0, ..., P_{max}\} \mathrm{W} \tag{2}$$

The probabilities (Equation (3)) and costs (Equation (4)) of each state transition can therefore be described as arrays **p** and **c**, respectively, to define a discrete-time Markov Decision Problem (MDP). These values are populated through analysis of the logged data and running the vehicle model, see Section 3.4.

$$p_{ijk} = \Pr(\mathbf{S}_{t+1} = j | \mathbf{S}_t = i, P_{FC,t} = k) \tag{3}$$

$$c_{ijk} = c(\mathbf{S}_{t+1} = j, \mathbf{S}_t = i, P_{FC,t} = k) \tag{4}$$

The total expected cost of performing an action, $k$, given the current vehicle state, $i$, is given by summing the product of each potential final state's ($j$) probability and cost:

$$\mathbf{C}(i,k) = \sum_{j=1}^{n} \left\{ p_{ijk} c_{ijk} \right\} \tag{5}$$

The optimal control strategy, $\pi^*(S)$, represents the set of actions for each state which will minimise the expected cost, $E$, over a given horizon. The Microcab is not intended to be charged externally during normal usage, therefore we assume a charge sustaining strategy with an infinite horizon. To ensure the policy converges, future time steps are discounted using a discount factor, $\alpha$. Therefore, the total expected cost, $J$, given the initial vehicle state, $\mathbf{S}_0$, and control policy, $\pi$, is calculated as follows:

$$J_\pi(\mathbf{S}_0) = \lim_{T \to \infty} E \left\{ \sum_{t=0}^{T-1} \alpha^{t-1} \mathbf{C}(\mathbf{S}_t, \pi(\mathbf{S}_t)) \right\} \tag{6}$$

The strategy which minimises the total cost is found using a policy iteration algorithm as described in [15]. This works by alternatively iterating policy evaluation and policy improvement steps until the policy converges. For this work, we used a discount factor, $\alpha$ of 0.9999, performed 100 policy evaluation steps for each improvement step and declared the policy to be converged when it remained constant for 36 policy improvement steps: representing 3600 s or 1 h of drive time. These values were chosen by trial and error to optimise computational efficiency whilst ensuring long-term charge sustaining behaviour.

## 3.4. Computational Efficiency

For each vehicle design, the model needs to be run approximately 2.5 million times, and therefore it is vital to produce the results as efficiently as possible. This was achieved by breaking the vehicle model down into three distinct stages, see Figure 3.

Stage one includes the vehicle drag, driveline and the electric motor, and produces a trace of electrical power draw required to run the input speed trace. This stage needs to be iterated once for each combination of speed and acceleration; a total of 105 iterations. The second stage calculates the fuel consumption and degradation for each combination of initial and final fuel cell power demand which are then used for the cost function. This stage also generates a trace corresponding to the electrical power generated by the fuel cell. This stage is independent of the vehicle speed and acceleration, and only needs to be simulated for each combination of current fuel cell power and fuel cell power demand, a total of 400 iterations in our case. The final stage uses the summation of the motor power and fuel cell power in order to calculate the response of the battery pack. As a result, it is only this part of the model which needs to be iterated for the full 2.5 million iterations.

To further increase efficiency, for each design, the results are stored whilst the sweep is taking place. This means that stages which have already been calculated for a previous vehicle design do

not need to be repeated if the results are valid for the current design. For example, stage 2 in Figure 2 is dependent only on the action and the maximum fuel cell power, and therefore independent of the battery size. These results can be re-used not only for each different initial battery SoC, but also for each individual battery capacity.

For the results presented below, the generation of the MDP probabilities and costs took around 18 h for each design and the SDP optimisation took approximately 6 h. A total of 169 designs were examined using around 4000 CPU-hours in total.

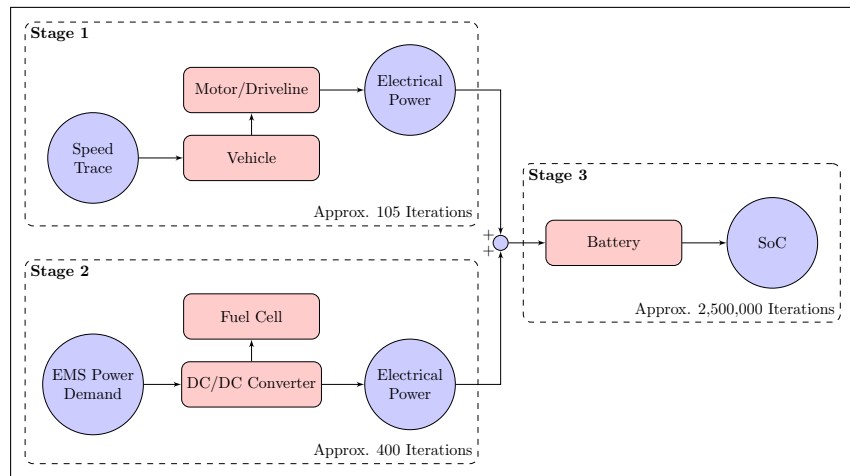

**Figure 3.** Flowchart of the simulation methodology.

## 4. Results

In order to assess its performance, each system design was simulated over 10 logged journeys.

### 4.1. System Performance and Feasibility

The result of each system design has been checked to ensure that it is capable of maintaining the battery SoC over all 10 logged journeys. The model is designed to stop simulating early if the battery voltage goes outside its validated range; therefore, any simulation which completes the full distance will produce valid results. It was found that all tested battery capacities were able to complete all the logged journeys without reaching the battery voltage limits, see Figure 4a. The minimum size fuel cell to meet the average power demand was dependent on the battery size. For small batteries ($\leq$1.8 kWh), a fuel cell of at least 1.8 kW is required for the vehicle to complete every cycle without an overall loss of battery SoC over a single journey. For vehicle designs with larger battery packs (>1.8 kWh), a fuel cell of at least 2.1 kW was required, see Figure 4b, due to the increased mass of the vehicle.

### 4.2. Fuel Consumption

The size of the fuel cell and battery pack will heavily affect the fuel consumption of the fuel cell. Figure 5 shows the calculated mean fuel consumption for each system design over these journeys. The minimum fuel consumption (12.6 g/km) is achieved using a 3.3 kW fuel cell and a 2.4 kWh battery pack. This represents an over-sizing of the original fuel cell by a factor of 2.75 and 12.5% increase in battery capacity and resulting in a 6.5% saving in fuel consumption despite the increased mass of the vehicle. The vehicle is fitted with 0.6 kg of hydrogen storage, giving a range of 47.4 km for the optimal design.

Above a certain fuel cell size, smaller battery capacities tend to result in a much higher fuel consumption. This is likely because the EMS is not able to control the battery voltage to within the specified limits without altering the demanded power from the fuel cell significantly. The smaller battery capacity directly means that there is a smaller energy buffer for transient loading and hence the fuel cell is required to run outside of its optimal efficiency more often. However, the issue is

compounded by the fact that the effective C-rate observed by the battery is higher, which results in a more significant voltage deviation for the same loading pattern. This reduces the effective capacity of the battery even further, resulting in significantly higher fuel consumption for undersized batteries. For larger battery packs than the optimum, the fuel consumption rises slightly. This can be attributed to the additional mass of the vehicle and shows that past a certain point, there is little benefit in using a higher capacity battery.

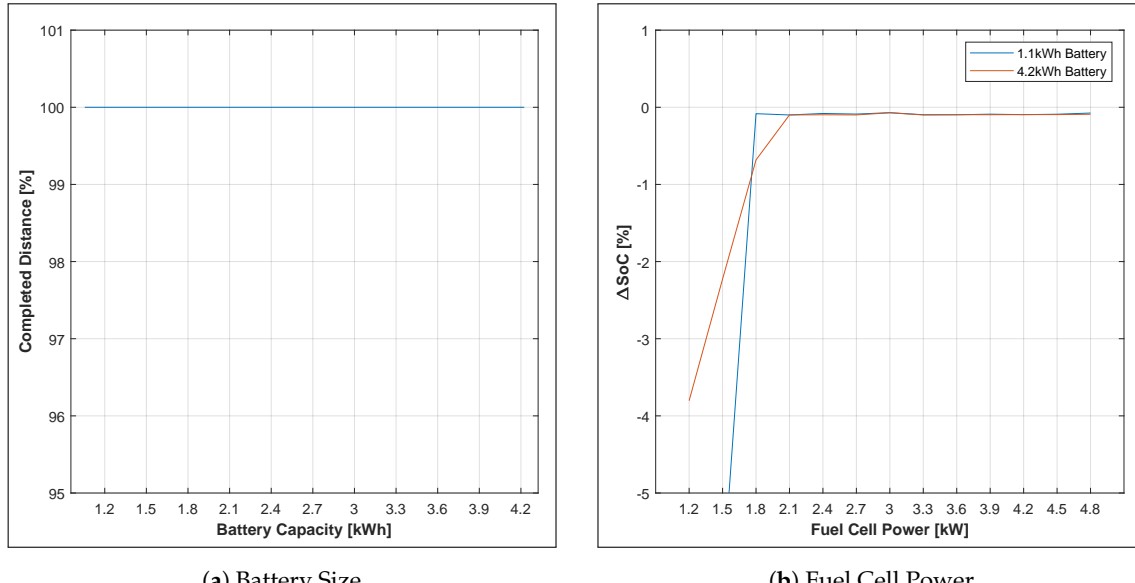

(**a**) Battery Size        (**b**) Fuel Cell Power

**Figure 4.** Feasibility study.

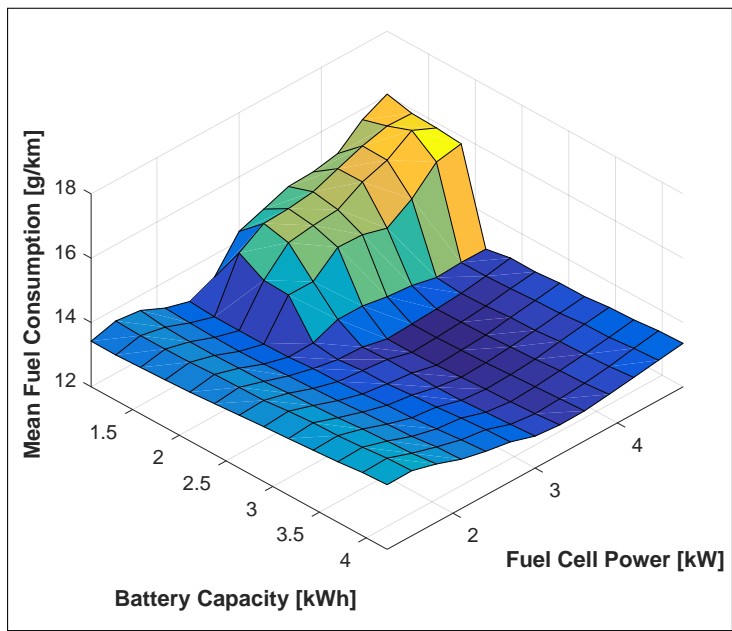

**Figure 5.** Fuel consumption.

Given a large enough battery capacity, the fuel consumption increases slightly when the fuel cell is either under or over its optimal size. This can be largely attributed to the shape of the operating efficiency curve of the fuel cell-DC/DC converter subsystem (Figure 6). Peak chemical conversion efficiency is around 37% at approximately 40% of its the maximum rated power output. At either side of this operating point, the efficiency drops slightly. For these system designs, the optimised EMS runs the fuel cell such that it tends to run at a constant average power demand, allowing the battery to

handle transient loading. This means that for optimal overall efficiency, the fuel cell size should be chosen such that the average power demand of the duty cycle coincides with fuel cell's peak efficiency.

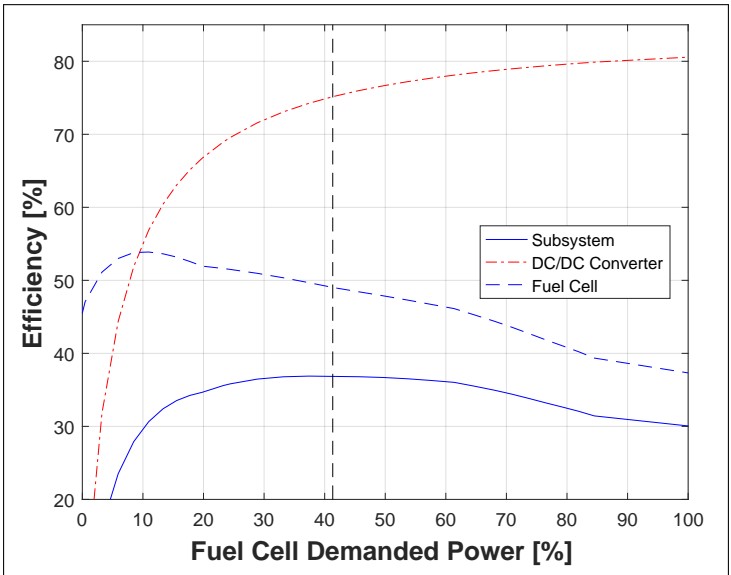

**Figure 6.** Fuel cell-DC/DC converter subsystem efficiency.

*4.3. Degradation*

Figure 7 shows the estimated fuel cell lifetime for each system design. Note that similarly to the fuel consumption, the best results are found with a fuel cell size at least 2.4 kW and a battery capacity of at least 2.4 kWh. In contrast to the fuel consumption, however, the degradation is significantly worse for systems designs not meeting these minimum criteria. For example, compared to the optimal fuel consumption design, a vehicle with a 3.3 kW fuel cell and a 1.1 kWh battery pack would experience a fuel consumption increase of 15%. This same design change would result in a 320% increase in fuel cell degradation, reducing its estimated lifetime from over 1200 h to just 374. In fact, the best design from a degradation point of view (3.3 kW, 4.2 kWh, 1259 h) would last approximately 3.4 times longer than the worst feasible design (4.2 kW, 1.8 kWh, 372 h).

This shows that although the objectives of minimizing fuel consumption and maximising fuel cell lifetime are largely complimentary [15], consideration of the fuel cell degradation is important when choosing a system design. For example, from a purely fuel consumption perspective, under-sizing the fuel cell slightly to reduce the cost of the design at the cost of a slight increase in fuel consumption might seem like a good compromise, however it will also significantly shorten the lifetime of the fuel cell.

For system designs using a fuel cell of at least 2.4 kW and a battery size of above 2.6 kWh, the lifetime of the fuel cell levels off. Only a marginal benefit is seen by increasing the size of the battery pack; consistently less than 4% by increasing from 2.6 kWh to 4.2 kWh. Similarly increasing the fuel cell size to 3.3 kW results in around 3% increase to lifetime over the 2.4 kW design, and above which a larger fuel cell is actually detrimental to the degradation rates.

The reason for this stabilisation is shown in Figure 8. For all fuel cell sizes, the on/off cycling dominates the degradation, representing up to 94% of the degradation seen. Further reduction of this type of degradation is impossible at the system level from either a design or control perspective because this represents just a single on/off cycle per journey. For smaller fuel cells, a combination of high relative loading and transient loading is responsible for additional degradation. As the maximum fuel cell power is increased, the degradation due to high relative loads and transient loading reduces. However, for fuel cells of 2.4 kW or higher, it can also be seen that increased frequency of operation at very low relative loads causes additional degradation which outweighs the benefits of the reduction in transient loading once the fuel cell power surpasses 3.3 kW.

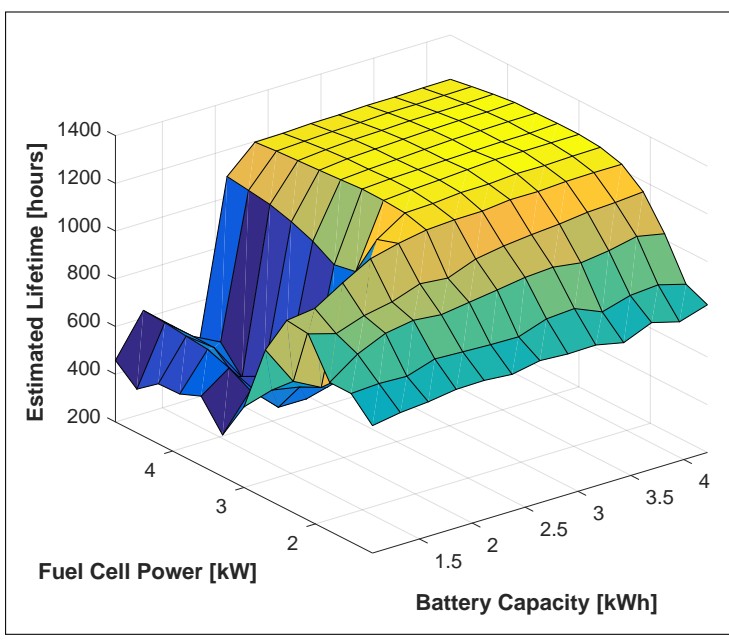

**Figure 7.** Lifetime (rotated for clarity).

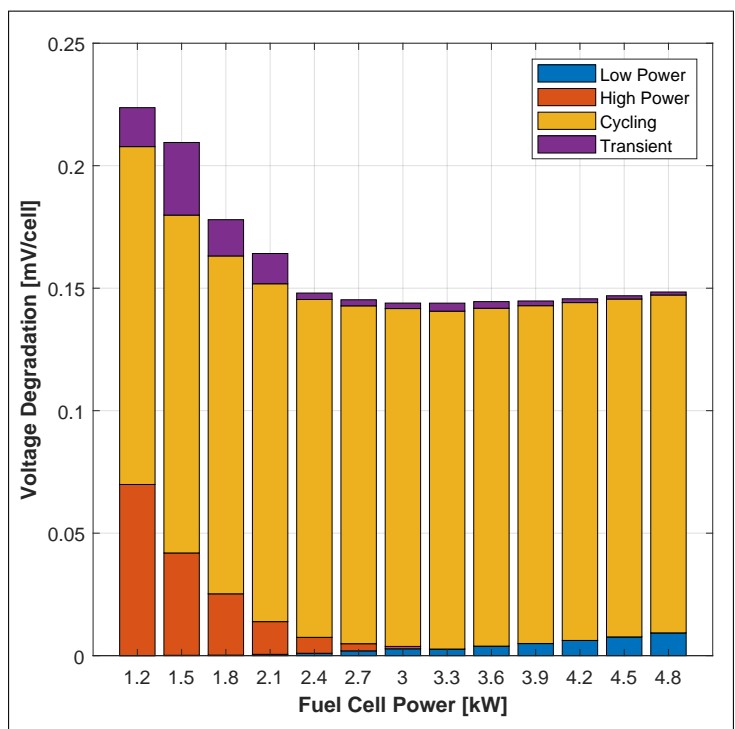

**Figure 8.** Breakdown of degradation rates (2.6 kWh battery).

*4.4. Operating Cost*

Figure 9 shows the overall average running cost of each system design. This represents a single metric which takes into account variations in fuel consumption, rates of degradation, vehicle mass and the cost of the fuel cell between the system designs and was also used to optimise the EMS. The running cost tends to follow similar trends to that of the efficiency and degradation, except with the added effect due to the increasing system cost which is proportional to the increased size of the fuel cell (assuming a fuel cell cost $50/kW).

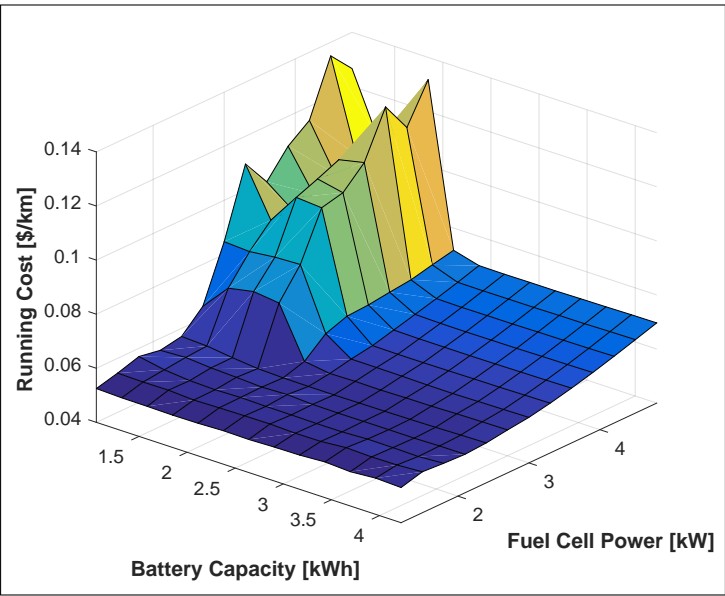

**Figure 9.** Running cost.

This shifts the optimal choice of fuel cell from 3.3 kW down to 2.4 kW (optimal cost). To show the reasons for this in more detail, Figure 10 shows a breakdown of the running cost as the fuel cell maximum power is varied for a battery capacity of 2.6 kWh. It can be seen that for smaller fuel cells, the cost of the fuel is the largest contributor to the change in overall running cost, but as the fuel size increases, the effectively fixed degradation associated with the single on/off cycle, combined with the increasing cost of the fuel cell, means that the small reductions in the fuel consumption and other types of degradation are outweighed by the additional proportional cost of the cycling degradation.

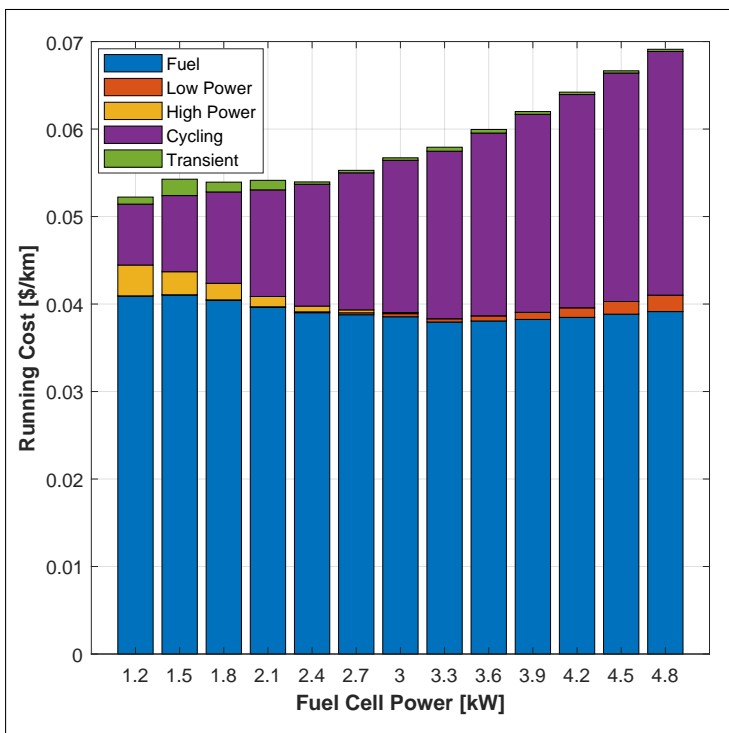

**Figure 10.** Breakdown of running cost using a 2.6 kWh battery (Note: 1.2 kW and 1.5 kW fuel cells were unable to complete the test cycles, see Figure 4b).

*4.5. Discussion*

It has been shown that the different objectives mentioned above result in slightly different optimal designs for the vehicle. The optimal design for the fuel consumption and for fuel cell lifetime use the same size fuel cell; however, the optimal lifetime design uses a significantly larger battery in order to reduce transient loading of the fuel cell. It has also been shown that the fuel cell lifetime is highly sensitive to the battery being undersized, whereas the fuel consumption is less so. However, the trade-off between these objectives should be analysed in more detail. Table 2 shows the comparison in performance of the optimal designs for each objective. It can be seen that the objectives are relatively complimentary, and designs optimised for one objective will still perform well against other objectives.

**Table 2.** Relative performance of each design.

|  | Optimal Fuel Consumption | Optimal Lifetime | Optimal Cost |
|---|---|---|---|
| Fuel Cell Power | 3.3 kW | 3.3 kW | 2.4 kW |
| Battery Capacity | 2.4 kWh | 4.2 kWh | 2.6 kWh |
| Fuel Consumption | 12.6 g/km | 12.9 g/km | 13.0 g/km |
| Relative Fuel Consumption | 100% | 102.2% | 102.8% |
| Range | 47.4 km | 46.4 km | 46.1 km |
| Relative Range | 100% | 97.9% | 97.3% |
| Lifetime | 1229 hr | 1259 hr | 1203 hr |
| Relative Lifetime | 97.7% | 100% | 95.5% |
| Cost | 0.0580 $/km | 0.0584 $/km | 0.0540 $/km |
| Relative Cost | 107.6% | 108.2% | 100% |

Figure 11 shows the trade-off between hydrogen consumption and fuel cell lifetime as a Pareto front. The Pareto front has a convex shape between the optimal lifetime and optimal fuel consumption designs, suggesting that compromise designs which weight these objectives against each other will be favourable over a design optimised for a single objective.

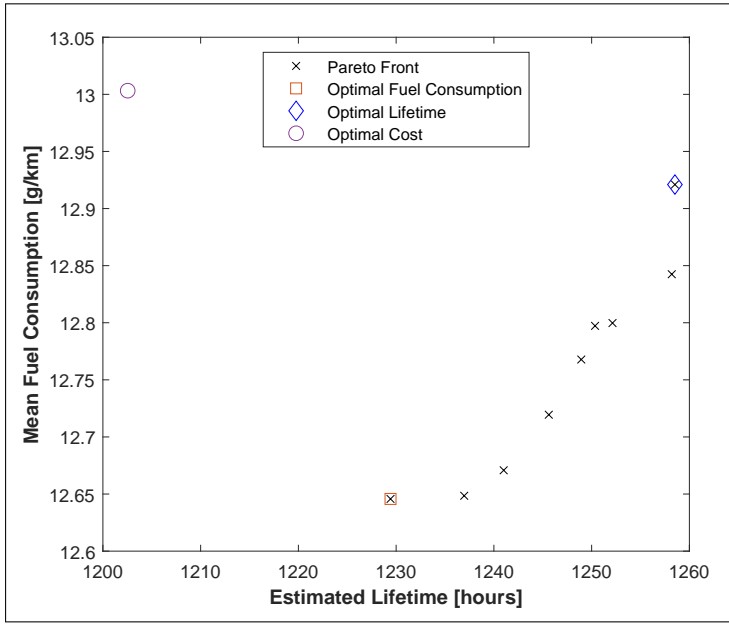

**Figure 11.** Pareto curve of fuel consumption vs. fuel cell lifetime.

Similarly, the optimal operating cost design uses a smaller fuel cell to offset the costs of degradation which are proportional to the fuel cell maximum power. Note that in Figure 11, the optimal cost design is clearly behind the Pareto front for hydrogen consumption and fuel cell lifetime. This is due to the increasing cost as the size of the fuel cell increases which is not taken into account in this comparison.

Figure 12 shows the Pareto fronts for range (Figure 12a) and lifetime (Figure 12b) against the operating cost. Note that in both cases, the Pareto front is convex allowing engineers to select a design which compromises the three criteria presented depending on the exact design requirements of the vehicle and the relative weighting applied to each objective.

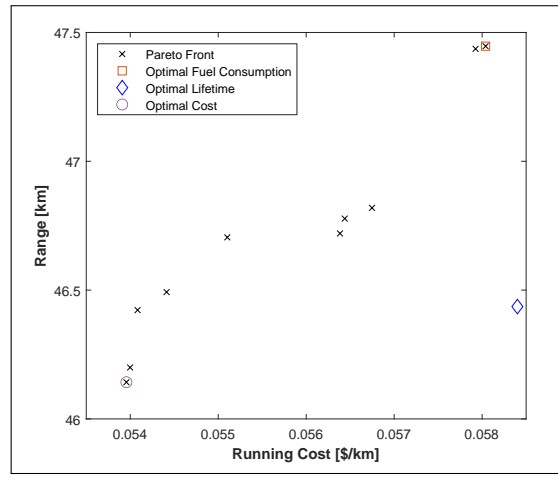
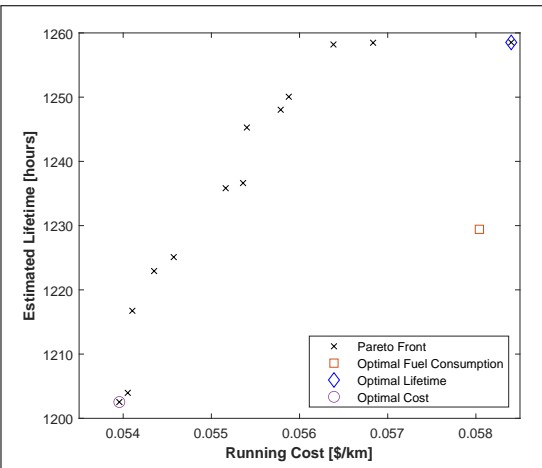

(**a**) Range vs. Operating Cost      (**b**) Stack Lifetime vs. Operating Cost

**Figure 12.** Pareto curve for overall operating cost.

## 5. Conclusions

This work has successfully demonstrated the effect that the fuel cell and battery sizes have on the fuel consumption, degradation, and overall running cost of FCHEVs. In particular, this work is believed to be the first to numerically demonstrate the benefits and drawbacks of attempting to extend the lifetime of the fuel cell stack through "over-sizing".

It has been shown that although using a stack sized such that the average power demand of the duty cycle coincides with peak efficiency of the fuel cell will minimise fuel consumption, the degradation associated with fuel cell start-up and shutdown, combined with the increased cost of a larger fuel cell, mean that it may be preferable to reduce the size of the fuel cell slightly to optimise the overall running cost of the vehicle. As the objectives show convex Pareto fronts when compared against each other, designs which represent a compromise between the objectives may be preferable depending on the exact design requirements of the vehicle, however the weighting of the objectives will depend on the manufacturer's preferences and the target market.

It has also been shown that significant improvement in the lifetime of the fuel cell can be made by increasing the fuel cell and battery capacity size when using an optimally controlled EMS. However, above a certain point for both components, the benefits level off and even fall slightly due to the increased mass of the vehicle. Therefore, care must be taken to choose an over-sizing factor which is high enough to realize these benefits without needlessly increasing the cost of the vehicle.

Finally, this work has concentrated on a particular case study with regard to the Microcab H4, which is a low-power campus-based vehicle. As a result, the numerical results are somewhat smaller than what would be expected for a full-size passenger vehicle. However, the techniques demonstrated are equally valid for examination of any type of vehicle and similar trends would be expected.

### Further Work

There are a number of ways in which these techniques could be extended. Of particular interest to the authors is the inclusion of a battery degradation model in order to assess the effects on the lifetime and running cost of the battery pack. The test vehicle currently uses lead acid batteries to minimise cost, and due to their low replacement cost compared to the fuel cell, their degradation has been, so far, neglected. However, lithium-ion battery technology has become more mature, and hence

less expensive, since the design of the vehicles. As a result, it may be useful to investigate the potential benefits (and drawbacks) of switching battery technology.

**Author Contributions:** Conceptualization, T.F.; methodology, T.F.; software, T.F.; investigation, T.F.; writing—original draft preparation, T.F.; writing—review and editing, K.E., supervision, K.E. All authors have read and agreed to the published version of the manuscript.

**Funding:** This research was funded by the Engineering and Physical Sciences Research Council (EPRSC) as part of the Doctoral Training Centre in Hydrogen, Fuel Cells and Their Applications (EP/G037116/1).

**Acknowledgments:** The authors would like to acknowledge Iain Staffell for providing the logged data of the test vehicles from their use on the University of Birmingham campus in order to build the stochastic drive-cycle model, Microcab Ltd. for providing the test vehicle, Martin Watkinson of HORIBA-MIRA and Rob Thring for their support and guidance during the preliminary stages of this work.

**Conflicts of Interest:** The authors declare no conflict of interest.

## Abbreviations

The following abbreviations are used in this manuscript:

| | |
|---|---|
| DDP | Deterministic Dynamic Programming |
| ECASA | Electro-Chemical Active Surface Area |
| EMS | Energy Management Strategy |
| FCHEV | Fuel Cell Hybrid Electric Vehicle |
| MDP | Markov Decision Problem |
| SDP | Stochastic Dynamic Programming |
| SoC | State of Charge |

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
