# Peer review of "The Effect of Fuel Cell and Battery Size on Efficiency and Cell Lifetime for an L7e Fuel Cell Hybrid Vehicle"

_energies, doi:10.3390/en13225889_

Round 1
Reviewer 1 Report
This research focused on components size optimization of a fuel cell hybrid vehicle. And base on the fuel consumption and component operation cost to evaluate the total running cost. This article is very interesting to read.
In Figure 9, the unit of running cost is ‘Range [km]’. Authors should include a description about the relation between Range and Running cost.
In this article, the cost of fuel cell and fuel is assumed to be $50/kW and $3/kg. The number is based on references 14、54、59、60. These references are relative old, and these number would not represent the current cost. Authors should replace these number with currently updated data, or include the information that this cost of fuel cell would be achieved in which year.
Author Response
Thank you for your comments, we have found them very useful in improving the quality of our publication. With reference to the relationship between range and running cost, a subsection (4.5, page 10-11) has been added to the results section to discuss the trade-offs between the fuel consumption, range, lifetime and running cost which includes a new comparison table of optimal results (Table 2) and Pareto curves (Figures 11 and 12) showing the non-dominated designs. A line has also been added to the conclusions section and the abstract has been updated to summarise these important new conclusions.
With regard to the cost assumptions, updated references (from 2018 and 2020) and further explanation are now provided in the modelling section on page 4, lines 154 to 162.
Reviewer 2 Report
The Effect of Fuel Cell and Battery Size on Efficiency and Cell Lifetime for an L7e Fuel Cell Hybrid Vehicle
The authors report an interesting strategy for modeling appropriate fuel cell and battery capacities for application in a fleet of forth Microcab H4s FCHEV. Different parameter as fuel consumption, degradation, and cost were considered and every system was modeled using a stochastic dynamic programing method. The method allowed to obtain an optimal design from prioritizing performance and costs demonstrating the benefits of properly designing of the power components. This method can represent an important strategy which has the potential to extrapolate to FCHEV’s of higher capacity.
In general, the structure of this manuscript is clear and well done. The experimental section and methodology are well described and clear. The discussion was thorough and well done, no typos or grammar mistakes were found.
The study is very relevant as a source of information for planning and designing in the fuel cell field.
The study recognizes the positive impact on reducing costs and increasing the lifetime of hybrid systems fuel cell-battery from an adequate design on their components, which contribute towards the massive application of these types of devices as alternative technology in the energy field.
I do recommend this article to be publish without further changes.
Author Response
Thank you for your thorough and positive review.
Reviewer 3 Report
Work of high interest and novelty due to the degradation modeling aspect.
Minor corrections :
- line 293 : "of its" instead of "of it's"
- line 315 : "plateaus" may be a difficult term for non english readers, "stabilises" may be prefered
Author Response
Thank you for your kind comments, further proofreading has been performed in order to fix spelling, grammar and typographical mistakes including three instances of “it’s” that have been replaced with “its”, “plateaus” has been replaced with “levels off” and “plateau” with “stabilisation” for clarity among non-native English readers.